# Structural mechanism of GTPase-powered ribosome-tRNA movement

Valentyn Petrychenko [1], Bee-Zen Peng [2], Ana C. de A. P. Schwarzer [1,3], Frank Peske [2], Marina V. Rodnina [2✉] & Niels Fischer [1✉]

GTPases are regulators of cell signaling acting as molecular switches. The translational GTPase EF-G stands out, as it uses GTP hydrolysis to generate force and promote the movement of the ribosome along the mRNA. The key unresolved question is how GTP hydrolysis drives molecular movement. Here, we visualize the GTPase-powered step of ongoing translocation by time-resolved cryo-EM. EF-G in the active GDP–Pi form stabilizes the rotated conformation of ribosomal subunits and induces twisting of the sarcin-ricin loop of the 23 S rRNA. Refolding of the GTPase switch regions upon Pi release initiates a large-scale rigid-body rotation of EF-G pivoting around the sarcin-ricin loop that facilitates back rotation of the ribosomal subunits and forward swiveling of the head domain of the small subunit, ultimately driving tRNA forward movement. The findings demonstrate how a GTPase orchestrates spontaneous thermal fluctuations of a large RNA-protein complex into force-generating molecular movement.

[1] Department of Structural Dynamics, Max Planck Institute for Biophysical Chemistry, Göttingen, Germany. [2] Department of Physical Biochemistry, Max Planck Institute for Biophysical Chemistry, Göttingen, Germany. [3]Present address: Department of Molecular Biology, University Medical Center Göttingen, Göttingen, Germany. ✉email: rodnina@mpibpc.mpg.de; niels.fischer@mpibpc.mpg.de

Each time the ribosome moves along the messenger RNA (mRNA) by one codon, the transfer RNAs (tRNAs) are displaced from the A (aminoacyl) to P (peptidyl) and from the P to E (exit) sites[1–3]. The reaction is facilitated by elongation factor G (EF-G), a translational GTPase. Time-resolved kinetic studies suggested that EF-G hydrolyses guanosine triphosphate (GTP) rapidly after binding to the ribosome[4], but the release of the reaction product inorganic phosphate (Pi) is delayed and coupled to rapid tRNA–mRNA movement[5,6]. During translocation, small and large ribosomal subunits (SSU and LSU) rotate relative to each other and the SSU head domain swivels relative to the SSU body[2,3]. These principal motions of the ribosome components are spontaneous and rapid even in the absence of EF-G[7–10]. The movement of the tRNA CCA ends on the LSU is spontaneous as well[2]. However, these thermal motions alone, in the absence of EF-G, are not sufficient to promote the displacement of tRNAs together with the mRNA on the SSU. tRNA movement on the SSU is the rate-limiting step of translocation in the absence of EF-G[11] and is slow in the absence of GTP hydrolysis[4,6,12]. Previous cryo-electron microscopy (EM) and X-ray structures showed EF-G in different states on the ribosome[13–20], but structural information on the early GTPase-driven steps is missing. How EF-G powered by GTP hydrolysis synchronizes the spontaneous fluctuations of the ribosome and the tRNAs into a rapid, directed motion is therefore the key unresolved question.

## Results

**Cryo-EM of early translocation states.** Structural elucidation of EF-G-dependent translocation is challenging because it takes place within milliseconds[4]. We visualized the early steps of translocation by time-resolved cryo-EM (Fig. 1 and Supplementary Fig. 1) using native *E. coli* pretranslocation complexes (PRE) and EF-G–GTP, but slowed down translocation by lowering the reaction temperature, by adding polyamines and the antibiotic apramycin (Apr)[21]. Apr has little effect on the early steps of translocation, such as GTP hydrolysis by EF-G or Pi release, but inhibits the late steps, in particular the completion of the tRNA movement from the A to P site (Fig. 2a and Supplementary Fig. 2a–d). For the cryo-EM experiment, we have chosen a time point on the translocation trajectory where GTP hydrolysis has occurred, but Pi release is delayed (Fig. 2a), which allowed us to visualize EF-G in its GDP–Pi- and GDP-bound states on translocating ribosomes (Fig. 1).

We obtained seven main structures by sorting cryo-EM particle images (Fig. 1a, Supplementary Fig. 1, Supplementary Tables 1 and 2, and Methods). The major population of PRE ribosomes does not contain EF-G and is found in the non-rotated state with tRNAs in their classical A/A and P/P positions, which we denote as classical (C) state (Fig. 1a, b); the preference for the C state is likely due to the presence of polyamines and Apr in the buffer. A smaller fraction of PRE complexes without EF-G is in a rotated state with tRNAs in hybrid (H) states, denoted H1 or H2 depending on the orientation of the A-site tRNA elbow region relative to the LSU. At 2.35 Å resolution, the structure of the C state shows the exact coordination of Apr in the SSU decoding center, including $Mg^{2+}$ ions and multiple water molecules (Fig. 2b). The differences to the earlier 3.5 Å structure of Apr on the isolated 30S subunit[22] may be relevant for antibiotic development because Apr is an attractive lead compound due to its insensitivity to common aminoglycoside resistance mechanisms and low ototoxicity in animal models[22,23]. Secondary Apr binding sites are located at the SSU shoulder and in the middle part of helix 44 (h44) of 16S ribosomal RNA (rRNA) (Supplementary Fig. 2e, f). Apr binding to the latter site stabilizes

the C state (Fig. 1a), explaining the observed effect of the antibiotic on subunit rotation[24].

Four structures show EF-G bound to the ribosome in early states of translocation (Fig. 1a). One of these classes contains clear density for GDP–Pi in the nucleotide-binding pocket of EF-G (Fig. 2c), showing the factor in the active GDP–Pi form bound to the PRE complex prior to SSU unlocking[5]. The ribosome is rotated and the tRNAs are found in H1 or H2 position, as in the state without EF-G bound (Fig. 1). In the lower resolution, H2–EF-G–GDP–Pi state, the nucleotide density would be compatible with either GTP or GDP–Pi (Supplementary Figure 1d). However, biochemical data showing that GTP has been hydrolyzed at this time point of cryo-EM preparation (Fig. 2a) and the overall structural similarity to the H1–EF-G–GDP–Pi complex suggests the presence of GDP–Pi, rather than GTP, also in the H2 state with EF-G. The present complex with EF-G–GDP–Pi differs dramatically from previously reported crystal structures of ribosome complexes with EF-G and non-hydrolyzable GTP analogs, GDPNP or GDPCP[13,15,17] (Supplementary Fig. 3a). Non-hydrolyzable GTP analogs do not block translocation completely, but rather slow it down by switching the ribosome to an alternative slow-gear translocation pathway[6,12]. In the crystal structures with EF-G-GDPCP/GDPNP, the ribosomes move into a POST state, with tRNAs in the P and E sites on the SSU or without tRNAs[13,15,17]. Accordingly, also the position of EF-G (Supplementary Fig. 3a) and the global ribosome conformations in these POST states are substantially different from our PRE–EF-G–GDP–Pi states, which present early translocation intermediates after EF-G engagement but before the tRNAs have moved.

Another two classes depict EF-G in the GDP-bound form with the tRNAs moved to the chimeric (CHI), or ap/P and pe/E states[14,20], with the tRNA anticodons remaining bound to the A- and P-site elements on the SSU head domain, but displaced to the P- and E-site positions on the SSU body[14,20]. Depending on the exact position of EF-G, the complexes are classified as CHI1 or CHI2 (Fig. 1a). Notably, we did not identify particles with EF-G in the GDP–Pi form and tRNAs in CHI states, consistent with the notion that Pi release and the onset of tRNA translocation are coupled[5]. However, we found a minor population of H1 complexes with EF-G in the GDP form, which represents ribosomes on which GTP hydrolysis and Pi release did not result in tRNA movement (Supplementary Fig. 4). The main function of EF-G is to unlock translocation on the SSU, the major energy barrier on the translocation pathway. Unlocking occurs after GTP hydrolysis upon Pi release by EF-G[5] and movement into the CHI state[14,20]. Thus, the present PRE–EF-G–GDP–Pi and the CHI–EF-G–GDP structures represent key intermediates at the onset of translocation that show how the chemical energy from GTP hydrolysis and Pi release translates into the forward movement of the mRNA–tRNA complex.

**EF-G in the active GDP–Pi state prior to translocation.** The binding of EF-G and GTP hydrolysis facilitates rotation of the ribosomal subunits and the concomitant tRNA movement from C to H[9,25–27] (Fig. 1b, c). The ratio of H1 to H2 on rotated ribosomes is independent of EF-G (Supplementary Fig. 1b), indicating that factor binding or GTP hydrolysis do not affect the tRNA dynamics on the LSU. In the nucleotide-binding pocket of EF-G, GTP coordination is similar to that in other EF-G structures with GTP analogs[13,15,17] (Fig. 3 and Supplementary Fig. 3a); a shift in the position of Pi compared to the γ-phosphate is consistent with the trajectory of the hydrolysis reaction. However, the structure of the key GTP sensor, the switch 1 region (sw1, residues 32–65), which is

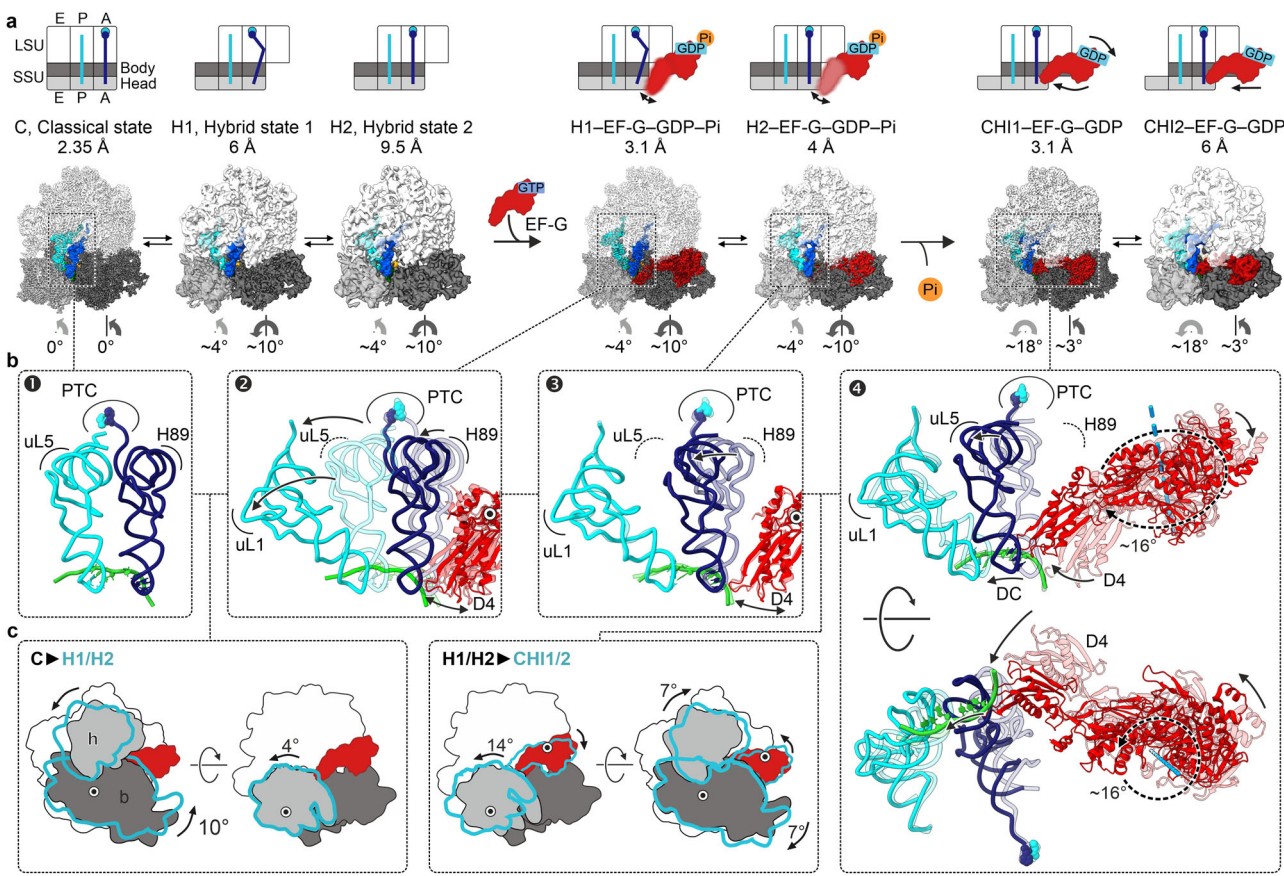

**Fig. 1 Snapshots of translocation upon GTP hydrolysis and Pi release by EF-G. a** Snapshots of early translocation captured by time-resolved cryo-EM. Schematics indicate tRNA positions (A, P, E), changes in the relative orientation of ribosomal subunits vs. C state (LSU, white; SSU body domain, dark gray; SSU head domain, light gray), and major movements of EF-G (black arrows). Cryo-EM maps of distinct states are shown with the respective resolution. PRE complex contains A-site fMet-Phe-tRNA$^{Phe}$ (blue) and P-site tRNA$^{fMet}$ (cyan). **b** Global changes in tRNA and EF-G positions. PTC peptidyl transferase center, H89 helix 89 of 23S rRNA; uL1, uL5 proteins of the LSU. The flexibility of EF-G domain 4 (D4) in H1 and H2 states is indicated by arrows (see Supplementary Fig. 5b, c for details). **c** Global changes in SSU and LSU conformations from C (white, gray, and black) to H1/H2 (blue contour) states (left panel) and from H1/H2 (white, gray, and black) to CHI1/2 (blue contour) states (right panel). Degrees of SSU body (b) and head (h) domain rotation are indicated by an arrow.

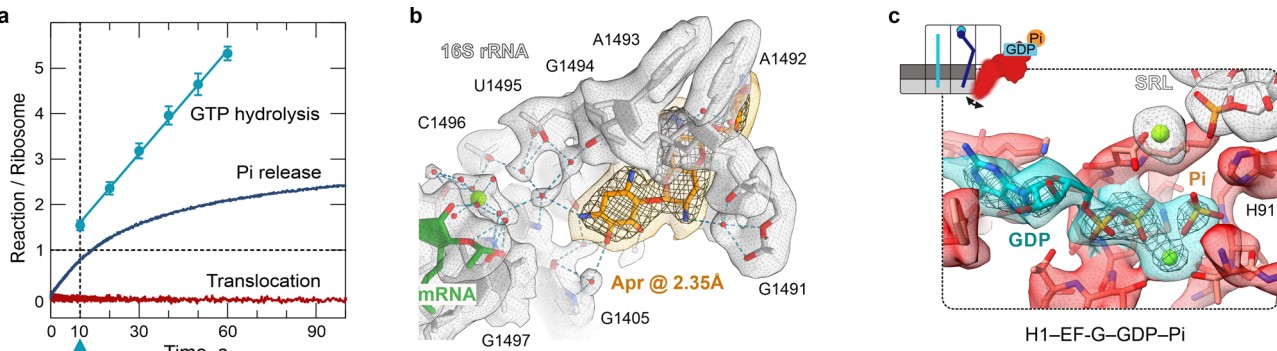

**Fig. 2 Timing of complex preparation and high-resolution features. a** Rapid kinetics analysis of translocation under cryo-EM conditions (4 °C, with Apr). The blue triangle marks the time point (10 s) of cryo-EM preparation (Methods). GTP hydrolysis: error bars represent the standard deviation of three independent experiments ($N = 3$). Pi release: time courses are averages of four technical replicates ($N = 4$). Translocation: time courses are averages of eight technical replicates ($N = 8$). **b** High-resolution structure of Apr bound to the SSU decoding center in C state. **c** GTP-binding pocket of EF-G in the GDP–Pi state. The cryo-EM density at a high threshold (5σ, wide mesh) shows the Pi as a separate, non-covalently coordinated entity; transparent surfaces, cryo-EM density at a lower threshold (3σ).

partially unfolded in other EF-G structures[13,15,17], is fully compacted in the EF-G–GDP–Pi complex (Fig. 3 and Supplementary Fig. 3b). The N-terminal part of sw1 adopts a compact fold that was not observed before, whereas the C-terminal part forms a short α-helix, as in posttranslocation complexes (POST) with GTP analogs[13,15,17] (Supplementary Fig. 3c). Importantly, the compact sw1 forms a connection between SSU and LSU, which appears to stabilize the intersubunit rotation.

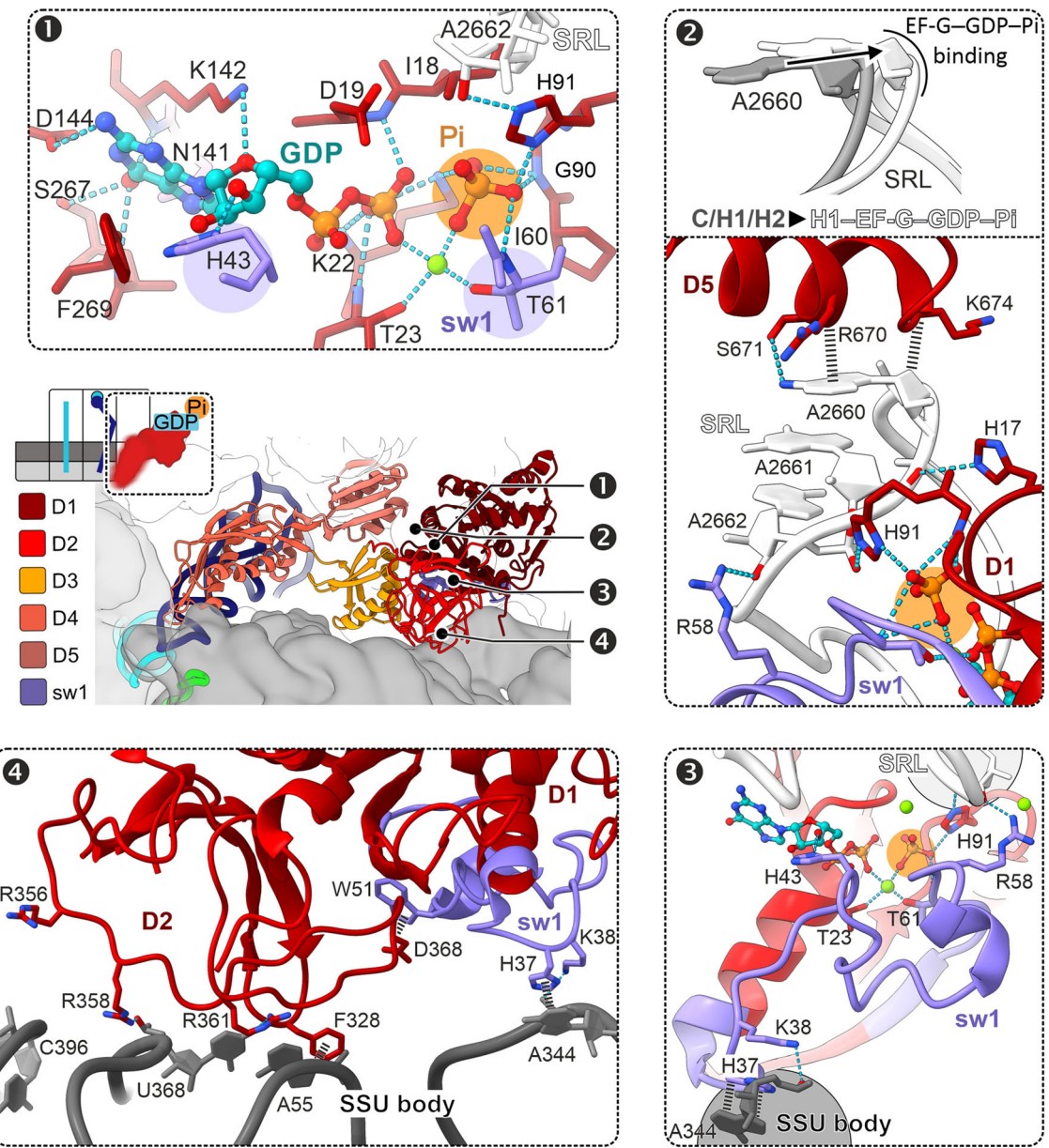

**Fig. 3 Structure of EF-G–GDP–Pi in H1 state.** Panels 1–4 depict distinct regions as indicated in the overview (middle left) with EF-G D1–D5 and sw1. Panel 1: Pi coordinates a network of interactions that stabilize the compact sw1 structure. Panel 2: interaction with EF-G–GDP–Pi induces a unique SRL twist. Panel 3: compacted sw1 bridges between LSU and SSU, thereby stabilizing the rotated state. Panel 4: interactions of EF-G D2 and sw1 with the rotated SSU body domain.

The compact fold of sw1 is supported by a network of interactions with the Pi and switch 2 (sw2) region (residues 86–105) of EF-G domain 1 (D1), the SRL of 23S rRNA on the LSU, and h14 of 16S rRNA in the SSU body domain (panels 1–3 in Fig. 3). Pi coordinates the invariant Thr61 of sw1, Gly90, and the catalytic His91 of sw2, and an $Mg^{2+}$ ion that also bridges with the β-phosphate of GDP. His43 of sw1 contacts the GDP ribose, while Arg58 of sw1 and His91 of sw2 interact with A2662 of the SRL. EF-G–GDP–Pi induces a unique twisted SRL conformation (panel 2 in Fig. 3 and Supplementary Fig. 3d). The SRL is essential for the GTPase activation of several translation factors, but the twisted conformation has not been described before (Supplementary Fig. 3d). The twist affects residues 2655–2665, with a maximum displacement at the apical part of the SRL where the bases A2660–A2662 move by >2.5 Å. On the SSU, EF-G sw1 binds to the SSU body domain, with His37 and Lys38 contacting A344 of h14. EF-G D2 contacts A55 of

h5 and U368 of h15 of 16S rRNA, and EF-G D3 interacts with protein uS12 of the SSU (panels 3 and 4 in Fig. 3 and Supplementary Fig. 5a). EF-G D4 points into the decoding center on the SSU, but appears flexible, as visualized by sorting the cryo-EM data into further substrates in which the tip of D4 moves from h34 at the SSU head towards the A-site tRNA (Supplementary Fig. 5b, c).

**How Pi release initiates tRNA movement**. Pi release triggers a large-scale repositioning of EF-G, global rearrangements of the ribosome, and the tRNA movement on the SSU (Figs. 1 and 4 and Supplementary Figs. 6–8). EF-G pivots around the SRL and rotates deeper into the cleft between the ribosomal subunits towards the decoding center, where it contacts the A-site tRNA and the SSU (Fig. 4a and Supplementary Fig. 6a). The conformation of the ribosome changes dramatically: the SSU body

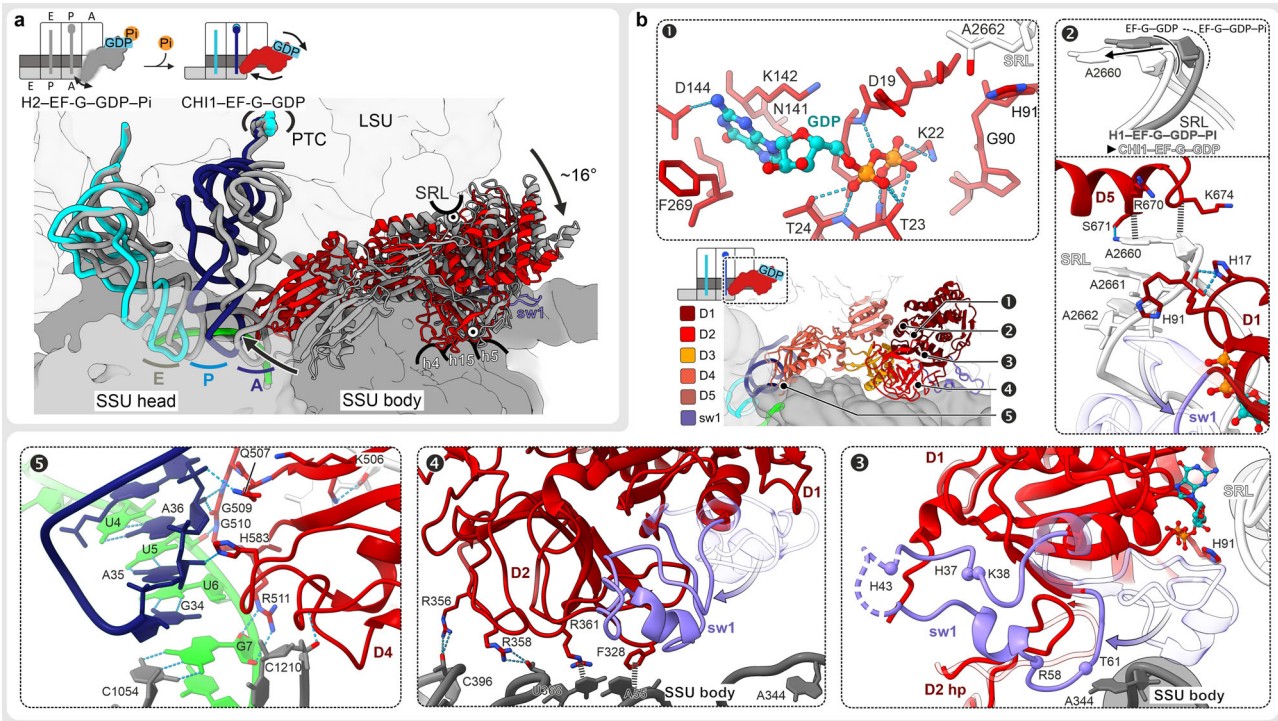

**Fig. 4 Large-scale rearrangements triggered by Pi release from EF-G. a** Rearrangements of EF-G and tRNAs from H2 (gray) to CHI1 state (colored); SSU domains and LSU are only shown for CHI1. Hinge points of rotation are indicated as white circles with a black dot. **b** Structure of EF-G–GDP in CHI1 and refolding of sw1 upon Pi release. For comparison, parts of EF-G–GDP–Pi in H1 state, aligned to EF-G D1, are included in semitransparent. Center: overview illustrating the regions shown in panels 1–5. Panel 1: nucleotide-binding pocket of EF-G–GDP on the ribosome with substantially fewer interactions as compared to the GDP–Pi state. Panel 2: sw1 loses interactions with the SRL resulting in relaxation of the SRL into its ground state conformation. Repositioning of EF-G also releases the interaction of sw2, i.e., H91, with the SRL. Panel 3: refolding of sw1 upon Pi release obliterates the connection that stabilized the rotated state, which results in the back-rotation of the SSU body. A hairpin (hp) structure of EF-G D2 changes with sw1 and stabilizes the refolded sw1. Panel 4: EF-G D2 forms new contacts with the SSU body domain in a non-rotated conformation. Panel 5: EF-G D4 forms multiple contacts with the SSU head domain and the mRNA–tRNA complex in the decoding center that stabilize SSU head swiveling and chimeric tRNAs states.

domain rotates backwards relative to the LSU, whereas the SSU head domain continues the swiveling motion in the forward direction (Fig. 1c). The tRNAs move from H into CHI states (Supplementary Fig. 7). On the LSU, the tRNA 3′ ends stay in the P and E site, while the elbow of the dipeptidyl-tRNA moves fully into the P site and interacts with L5.

The loss of Pi coordination remodels the GTPase center, which, in turn, drives EF-G rotation (Fig. 4 and Supplementary Fig. 6a–c). The $Mg^{2+}$ ion dissociates, Lys22 now contacts the β-phosphate of GDP, Thr61, and Arg58 of sw1, and the catalytic His91 of sw2 shifts away from the SRL. The entire sw1 moves away from the SRL and refolds from the compact into an extended conformation (Supplementary Fig. 8a). A hairpin in D2 (residues 362–373) rearranges and stabilizes the extended conformation of sw1. EF-G D1 and D5 remain docked on the apical part of the SRL (Fig. 4b). However, the disruption of interactions with sw1 and sw2 allows the SRL to relax into its ground conformation. EF-G moves together with the SRL, resulting in a large-scale reorientation of EF-G on the ribosome (Fig. 4 and Supplementary Fig. 6a–c). Refolding of sw1 also disrupts its interactions with the SSU body, which now rotates backwards into the non-rotated state. EF-G D2 "rolls" over the SSU shoulder and forms additional interactions with h4 of 16S rRNA, thereby promoting back-rotation of the SSU (panel 4 in Fig. 4b and Supplementary Fig. 8b), while EF-G D3 accommodates to the changes by shifting its contacts on protein uS12 (Supplementary Fig. 5a).

As a result of EF-G rotation pivoting at the SRL, D4 swings into the decoding center, where it interacts with the tRNA–mRNA complex, the LSU (H69 of 23S rRNA), and the SSU head domain (panel 5 in Fig. 4b and Supplementary Figs. 7, 8d). Consistent with an earlier report[14], the conserved His583 and Gln507 of EF-G, which are involved in reading-frame maintenance[28,29], interact with the backbone of the A-site tRNA. As the tRNA–mRNA complex moves out of the A site on the SSU body domain, the triple helix with the universally conserved bases G530, A1492, and A1493 of 16S rRNA is resolved, but the codon–anticodon duplex is maintained and stabilized by multiple interactions with EF-G D4. The overall domain arrangement of EF-G in the GDP-bound state is sampled already in the EF-G–GDP–Pi state (Supplementary Fig. 6b), whereas the domain dynamics change (Supplementary Fig. 6d). Prior to Pi release, EF-G D1 and D2 are stably bound to the ribosome, whereas D4 is flexible. After Pi release, D4 is tightly bound in the decoding center, but D1 and D2 become more dynamic and move even further into the intersubunit space upon transition from CHI1 to CHI2 (Supplementary Fig. 8c), indicating the direction of EF-G movement when tRNAs move from CHI to the POST state.

## Discussion

The present structures show how EF-G utilizes GTP hydrolysis to synchronize spontaneous fluctuations of the ribosome to promote rapid forward tRNA–mRNA movement. The main effect of EF-G binding followed by GTP cleavage is to stabilize the rotated-hybrid state[25] of the PRE complex via a compact sw1, which links the rotated subunits and induces a unique SRL twist. The subsequent Pi release remodels the contacts of EF-G sw1 and sw2,

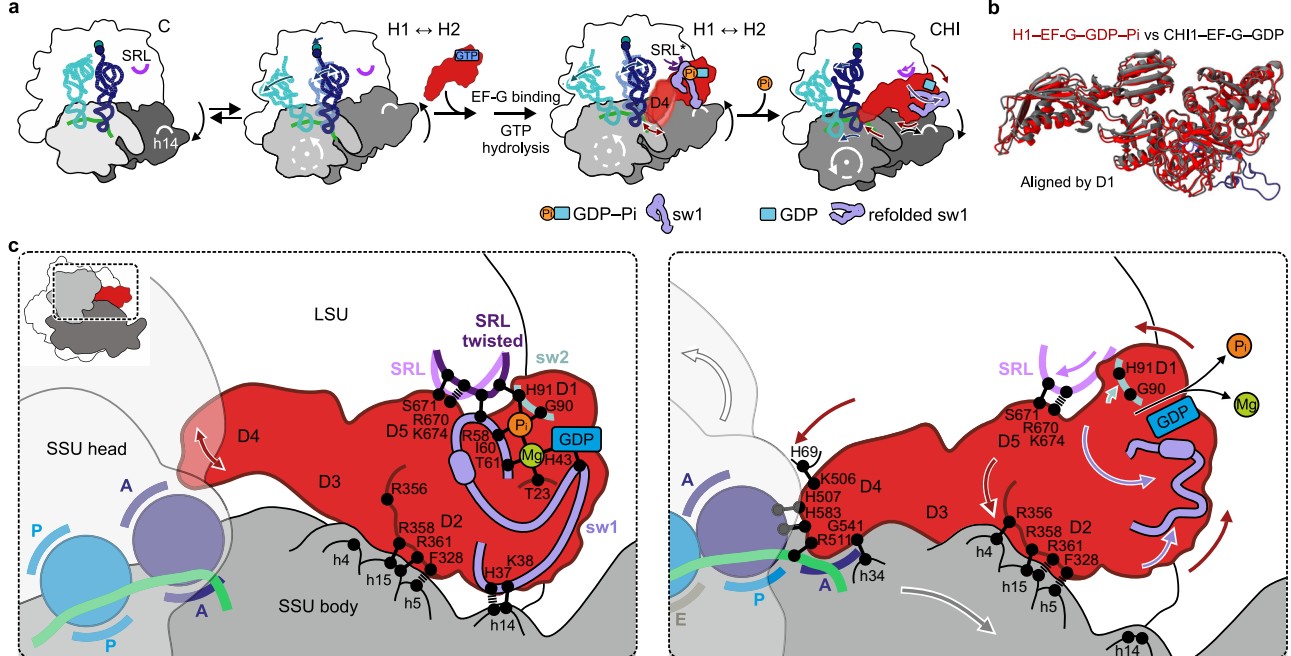

**Fig. 5 Coupling of GTP hydrolysis by EF-G to ribosome–tRNA movements. a** Overview of early translocation events as seen by cryo-EM. **b** Superposition of EF-G in GDP-Pi vs. GDP state. Note the similar overall conformation. **c** A GTPase "loaded spring" mechanism driving translocation. Left: binding of EF-G–GDP-Pi stabilizes a unique twist of the SRL via sw1 and sw2 of EF-G. The compact sw1 bridges between the SRL and h14 of 16S rRNA and stabilizes the SSU in the rotated state, whereas EF-G D2 forms numerous contacts with the SSU body domain (h5, h15). Fluctuations of EF-G D4 reach deep into the decoding site of the SSU (red arrow). Right: Pi release results in refolding of sw1 (purple arrows), which loosens the interactions that held the SSU in the rotated state and initiates a cascade of coupled structural changes: (i) the SSU body rotates backwards (gray arrow). (ii) The SRL relaxes back into its ground state (magenta arrow). (iii) EF-G rotates together with the SRL and rolls on the SSU shoulder (red arrows), forming new interactions with h4 that stabilize the non-rotated conformation of the SSU body domain. (iv) EF-G D4 binds into the decoding center of the SSU, stabilizing the SSU head domain in the swiveled state (white arrow) and promotes the movement of the tRNA–mRNA complex relative to the SSU into the chimeric tRNA sites.

which initiates a cascade of rearrangements and a large-scale reorientation of EF-G that "unlocks" the ribosome and ultimately promotes tRNA movement (Fig. 5 and Supplementary Movie 1). Unlocking overcomes the major barrier on the translocation pathway[5]. Our results are consistent with the single-molecule study suggesting that a ~10° (16° in our structures) global rotational motion of EF-G relative to the ribosome after GTP hydrolysis exerts a force to unlock the ribosome[30]. Our structural mechanism is further supported by the kinetic data that suggest a very similar choreography of SSU body and head domain rotation and tRNA–mRNA translocation in the absence of Apr[12]. In the absence of the antibiotic, the subsequent steps of translocation are rapid, enhanced by further movement of EF-G into the A site[6,12].

While this manuscript was under consideration, two reports appeared that also visualize early states of translocation by cryo-EM, albeit at somewhat different conditions. Carbone et al.[31] used time-resolved cryo-EM to follow translocation with EF-G–GTP at low temperature and in the presence of polyamines, similar to the approach used in the present paper, but without antibiotics and over a longer time scale. Their structures of early translocation states show PRE–EF-G–GDP–Pi (state III in that paper) and CHI–EF-G–GDP complexes (state IV) that are very similar to those described here. Minor differences, such as the tRNA positions in the PRE–EF-G–GDP–Pi complex, which are found exclusively in the H2 state in Carbone et al. but sample both H1 and H2 states in our structures, may be explained, e.g., by the different dipeptidyl-tRNAs in the A site. The CHI–EF-G–GDP structures appear to be practically identical in the two reports, except for the sw1 region, which is not resolved in Carbone et al. In addition, Carbone et al. observe late translocation states, including a POST–EF-G–GDP intermediate that has not been

captured before. Our and their data collectively support the notion that Pi release from EF-G triggers the tRNA movement to CHI states, which shows how GTP hydrolysis contributes to tRNA movement. The following rearrangements—also initiated by Pi release—result in the destabilization of EF-G binding, consistent with the dual role of GTP hydrolysis by EF-G in facilitating tRNA movement and EF-G-release at the end of translocation[32].

The second study reports on PRE–EF-G complexes in the presence of two antibiotics, spectinomycin (Spc) and fusidic acid (FA) (in Rundlet et al.[33]). The structure with EF-G–GDP-FA (INT2 in Rundlet et al.) is similar to the CHI–EF-G–GDP complex in this study and Carbone et al.[31], except for sw1, which is not resolved in Rundlet et al. and Carbone et al. structures. However, their PRE–EF-G–GTP complex stalled by Spc (INT1)[33] differs from our and the Carbone et al. structures. It shows EF-G in complex with GTP, which is surprising because GTP hydrolysis is very rapid even at low temperatures (Fig. 2a) and is not inhibited by Spc[34]. The mRNA–tRNA complex in their INT1–EF-G–GTP intermediate is partially unlocked, while the A-site peptidyl-tRNA and D4 of EF-G-GTP move to a position halfway between the PRE–EF-G–GDP–Pi and the CHI–EF-G–GDP complexes. The degree of subunit rotation in the Spc-stabilized PRE–EF-G–GTP[33] is similar to that in PRE–EF-G–GDP–Pi (this study and ref. [31]), but the head swiveling is more pronounced (7.5° vs. 4°). The INT1–EF-G–GTP complex by Rundlet et al. may represent a translocation intermediate halfway between PRE–EF-G–GDP–Pi and the CHI–EF-G–GDP complexes. Alternatively, the altered ribosome conformation may be caused by Spc, which is known to affect the dynamics of the SSU head swiveling[35,36], as well as backward rotation of the SSU

body[36]. The alterations in ribosome dynamics by Spc shift the ribosome to a slow gear[36,37] also seen with non-hydrolyzable GTP analogs[12]. The partial disruptions of the A-site mRNA–tRNA interactions with the SSU observed by Rundlet et al., which they interpreted in terms of EF-G-induced unlocking, may also arise from the Spc-induced destabilization of the A-site tRNA binding[34]. Accordingly, INT1–EF-G–GTP in Rundlet et al. may represent an Spc-induced state, rather than an authentic translocation intermediate.

GTPases use GTP hydrolysis to control their interactions with their downstream effectors by a "loaded spring" mechanism where the release of Pi after GTP hydrolysis allows the two switch regions to relax into the GDP-specific conformation[38]. In multidomain GTPases, relaxation of the "loaded spring" leads to a change in interdomain contacts of the GTPase domains often leading to global domain rearrangements. One example of a classical translational GTPase is EF-Tu, the factor that in its GTP-bound form delivers aminoacyl-tRNA to the ribosome. EF-Tu GTPase is activated by correct codon recognition by aminoacyl-tRNA. Upon Pi release, EF-Tu sw1 refolds into a β-hairpin, which changes contacts with D3 of the factor and facilitates its dissociation from aminoacyl-tRNA and the ribosome[39] (Supplementary Fig. 9). In contrast, in EF-G sw1 refolds into an extended structure underneath D1 and D2 and does not change the domain arrangement, but triggers a large-scale rigid-body rotation of EF-G, which acts to bias thermal motions of the ribosome–tRNA complex into the forward movement generating ~13 pN of force[40]. In summary, the present work shows how Pi-dependent remodeling of sw1 and sw2 regions is converted into rapid directed movement in a molecular machine essential for cellular function.

## Methods

**Complex preparation**. Preparation of PRE complexes and kinetic experiments were carried out in buffer A (50 mM HEPES (pH 7.5 at room temperature), 70 mM NH$_4$Cl, 30 mM KCl, 3.5 mM MgCl$_2$, and, where indicated, 0.6 mM spermine and 0.4 mM spermidine (polyamines, P). Ribosomes from E. coli (MRE600), f[$^3$H]Met-tRNA$^{fMet}$, [$^{14}$C]Phe-tRNA$^{Phe}$, and [$^{14}$C]Lys-tRNA$^{Lys}$ from total tRNA from E. coli MRE600, initiation factors, EF-Tu, and EF-G were expressed in BL21(DE3) and prepared according to standard protocols[32,41–43].

To prepare initiation complex (IC), 70S ribosomes were incubated with a 2-fold excess of mRNA, 1.7-fold excess of initiation factors, 3-fold excess of f[$^3$H]Met-tRNA$^{fMet}$, and 1 mM GTP in buffer A for 30 min at 37 °C. Ternary complex (TC) was prepared by incubating EF-Tu (3-fold excess over tRNA) with 1 mM GTP, 3 mM phosphoenolpyruvate, and 0.5% pyruvate kinase in buffer A for 15 min at 37 °C and subsequent addition of aminoacyl-tRNAs. PRE complex was formed by mixing IC and TC (2-fold excess over IC). Purification of IC and PRE was performed by centrifugation through a 1.1 M sucrose cushion in buffer A with 21 mM MgCl$_2$ for 2 h at 259,000 × g. Pellets were dissolved in buffer A with 21 mM MgCl$_2$, and the concentration of purified complex was determined by nitrocellulose filtration and radioactivity counting. Prior to cryo-EM grid preparation, traces of sucrose were removed from PRE preparation by buffer exchange into buffer A supplemented with Apr (50 μM) using Zeba Spin desalting columns (7 K MWCO, Thermo Fisher).

**Biochemical and rapid kinetics analysis**. mRNA translocation was measured in stopped-flow apparatus by monitoring changes in fluorescein fluorescence excited at 470 nm and after passing a KV500 cut-off filter (Schott)[34]; for collection of all stopped-flow data, the software Pro-Data SX 2.5.1852.0 (Applied Photophysics) was used. mMF+14Flu-programmed PRE (0.05 μM final) were rapidly mixed with EF-G (2 μM) in buffer A and 1 mM GTP at 4 °C and the signal change was recorded for 100 s. Several technical replicates (6–8) were collected and data averaged; the experiment was repeated twice.

Pi release was measured by monitoring the fluorescence change of phosphate-binding protein (PBP) labeled with MDCC (7-diethylamino-3-[[[(2-maleimidyl)ethyl]amino]carbonyl]coumarin)[44], excited at 425 nm and after passing a KV450 cut-off filter (Schott). PRE complexes (0.8 μM final) were rapidly mixed with EF-G (2 μM) in buffer A with MDCC-labeled PBP (2.5 μM) and GTP (25 mM) at 4 °C. To minimize phosphate contaminations, all solutions and the stopped-flow apparatus were treated with 0.1 mM 7-methylguanosine and 0.1 U/ml nucleoside phosphorylase[44]. Time courses are averages of four technical replicates (N = 4); the experiments were repeated twice.

The GTPase activity of EF-G was tested by incubating PRE (0.8 μM) and EF-G (2 μM) together with GTP (25 μM) with a trace amount of [γ-$^{32}$P]GTP in buffer A with polyamines at 4 °C. Reactions were quenched by adding an equal volume of 40% formic acid. Samples were analyzed via thin-layer chromatography (Polygram CEL 300, Macherey-Nagel) using 0.5 M potassium phosphate (pH 3.5) as the mobile phase[45]. The radioactivity was detected by phosphor imaging[45]. Error bars represent the standard deviation of three independent experiments (N = 3).

Tripeptide formation was measured by mixing IC (0.1 μM) programmed with mRNA coding for fMet, Lys, and Phe (mMKF), TC([$^{14}$C]Lys-tRNA$^{Lys}$) and TC(Phe-tRNA$^{Phe}$) (0.2 μM each), and EF-G (2 μM) in buffer A at 4 °C. At given time points, reactions were quenched with 0.1 volume KOH (1 M). Samples were hydrolyzed for 30 min at 37 °C, neutralized with 0.1 volume glacial acetic acid, and centrifuged. Peptides in the supernatant were separated by high-performance liquid chromatography on an RP-8 column and quantified by radioactive counting using the software QuantaSmart 4.00 (PerkinElmer). Error bars represent the standard deviation of three independent experiments (N = 3).

**Cryo-EM analysis**. Cryo-EM grids were prepared in a time-resolved manner at 4 °C with a total time of 10 s to vitrification after mixing of EF-G–GTP with PRE complex. Three microliters of PRE were mixed with 3 μl of EF-G–GTP in buffer A supplemented with 50 μM Apr, polyamines, and 1 mM GTP, resulting in final concentrations of 0.8 μM PRE and 2 μM EF-G. Ribosome–PRE complexes were applied to EM grids (Quantifoil 3.5/1 μm, Jena Bioscience) covered with pre-floated continuous carbon, manually blotted with filter paper (Whatman #1), and plunge frozen at 4 °C and 95% humidity using a custom-made device.

Cryo-EM data were acquired using a Falcon III direct electron detector (Thermo Fisher Eindhoven) at 300 kV acceleration voltage on a Titan Krios G1 microscope (Thermo Fisher Eindhoven) equipped with an XFEG electron source and a spherical aberration corrector (CEOS Heidelberg) that was tuned with the software CETCORPLUS 4.6.9 (CEOS Heidelberg). In total, 9300 cryo-EM movie images (4096 × 4096 pixels) were recorded in integration mode using EPU 2.3 (Thermo Fisher Eindhoven) with an exposure time of 1.09 s, a total dose of 30 e/Å$^2$, 20 fractions per movie, and a defocus range of 0.2–1.5 μm.

Cryo-EM movie images were motion-corrected using the MotionCor2[46] implementation in RELION 3.1[47], CTF parameters were estimated with CTFFIND-4.1[48], micrographs showing Thon rings worse than 3.5 Å were excluded and particles were selected using GAUTOMATCH 0.56 (K. Zhang, MRC-LMB, Cambridge). Subsequent cryo-EM image processing was performed in RELION 3.1[47]. Selected particles were sorted at 4.64 Å pixel size for particle quality by 2D classification, resulting in 1,326,729 ribosome particle images. 3D classification for particle quality and global ribosome conformation yielded two populations, with rotated and non-rotated subunits (step 1 in Supplementary Fig. 1e). Further sorting was carried out separately for the two groups at the final pixel size of 1.16 Å. Per-particle motion correction was performed using the Bayesian polishing approach[49] followed by CTF refinement[50] (per-particle defocus, per-micrograph astigmatism), and another round of Bayesian polishing and then CTF refinement on a 3 × 3 grid to account for off-axial aberrations (first magnification, then per-particle defocus, per-micrograph astigmatism, beamtilt, trefoil, and fourth-order aberrations); the default priors from RELION were used for both Bayesian polishing steps. As the large population of non-rotated ribosomes enabled very high-resolution refinement (to Nyquist frequency) and consequently accurate estimation of higher-order aberrations, the parameters estimated for this group were also used for the smaller population of rotated ribosomes, i.e., magnification, beamtilt, trefoil, and fourth-order aberrations.

The group of non-rotated ribosomes was sorted according to global ribosome conformation and data quality (step 2), excluding ribosomes with slightly different intersubunit rotation, resulting in the final homogeneous population of ribosomes in the C state. We also tried to identify non-rotated ribosomes with EF-G bound in this large group using focused classification with signal subtraction with various parameters and masks on different areas. We could not detect EF-G on classic pretranslocation state ribosomes, as expected given that EF-G binding facilitates[25] and stabilizes[9,25,51] ribosome rotation and tRNA movement into the hybrid states.

The population of rotated ribosomes was initially sorted for EF-G occupancy and orientation by focusing with a mask on D1 and D2 of EF-G (step 3), resulting in three major populations: (i) ribosomes with tRNAs in hybrid states, but without EF-G; (ii) ribosome–EF-G complexes with tRNAs in hybrid states; and (iii) ribosome–EF-G complexes with tRNAs in CHI states. All subsequent sorting steps were performed using focused classification with signal subtraction. The ribosomes without EF-G were classified according to A-site tRNA occupancy and position (step 4), which resulted in two final populations with ribosomes in the H1 and H2 states. The ribosome–EF-G complexes with tRNAs in hybrid states were sorted by D1 and D2 of EF-G (step 5), resulting in one population with density for EF-G–GDP–Pi, and a second population with EF-G in a different orientation on the ribosome. The latter group was complemented in step 7 by particles first assigned to the CHI states, but later shown to carry hybrid state tRNAs (step 6). The density for EF-G was improved in step 8 by another round of sorting for D1 and D2 yielding the final population of H1–EF-G–GDP complexes. Various attempts to improve the definition of EF-G D4 were not successful, suggesting that D4 is highly flexible in this complex. The complexes with EF-G–GDP–Pi were first sorted for the position of the A-site tRNA into H1 and H2 states with EF-G bound (step 9).

The ribosome–EF-G complexes in the H1 state were classified for D1 and D2 to obtain the final H1–EF-G–GDP-Pi population with well-defined EF-G density except for D4 (step 10). Further sorting on D4 in step 11, enabled visualization of three substates (1–3) of D4 that differed in the orientation of D4. Particles of the H2–EF-G–GDP–Pi group similarly showed an overall well-defined EF-G density, but very scattered density for D4. Sorting on the full EF-G resulted in one substate with improved density for EF-G (step 12). Ribosome–EF-G complexes with tRNAs in CHI states were first classified globally without a mask (step 6), identifying a minor group of H1 complexes with EF-G–GDP that was combined into the H1–EF-G–GDP population (see above) and a major group of ribosome–EF-G complexes with CHI tRNAs. The latter were then sorted for the complete EF-G (step 13), resulting in two groups that differed in the position of D1 and D2 of EF-G, the final population of the CHI1–EF-G–GDP and CHI2–EF-G–GDP complexes.

All final particle populations were refined to high resolution following the gold-standard procedure and the overall resolution of each reconstruction was determined using soft masks to exclude solvent (Supplementary Fig 1j). Global amplitude sharpening was performed in PHENIX 1.16-3549[52] with low-pass filtering to the respective final resolutions. Prior to sharpening, maps at <3.5 Å resolution were supersampled to 0.6525 Å per pixel in a 512[3] pixels box.

**Atomic model refinement.** An initial atomic model was created by the rigid-body fitting of atomic coordinates into the 2.35 Å cryo-EM map of state C with ChimeraX 1.2[53] using the following models: *E. coli* 70S ribosome and tRNA[Phe] from PDB 6YSS[54], tRNA[fMet] from 5LZD[55], Apr from PDB 4AQY[22]; mRNA and fMet-Phe dipeptide bound to tRNA[Phe] were modeled manually. An initial model for EF-G was built based on refinement of PDB 3J9Z[56] into the cryo-EM map of the H1–EF-G–GDP-Pi state. Generally, atomic models were partially refined manually in Coot 0.8.9.2–0.9.3[57] and then automatically by real-space refinement in PHE-NIX 1.16-3549[52] using phenix.real_space_refine to the final resolutions with global minimization, simulated annealing, atomic displacement parameters, and local grid search for seven macrocycles over 500 iterations each. Metal coordination and ligand restraints were prepared based on manually optimized initial models using phenix.ready_set. For the 2.35 Å cryo-EM map of state C, only well-resolved water molecules evident in both half maps were built around the Apr binding site. The H2–EF-G–GDP–Pi state and the lower resolution substates 1–3 of H1–EF-G–GDP-Pi were modeled based on the final model of the H1–EF-G–GDP–Pi state, for the latter EF-G D4 was refined in Coot using additional all-atom restraints. For the CHI1–EF-G–GDP state, the extended sw1 of EF-G was built based on the cryo-EM map low-pass filtered to 5 Å using iSOLDE 1.1.0[58] and Coot 0.9.3 with alpha-helix restraints for the residues 52–57. The CHI2–EF-G–GDP state was modeled starting with the CHI1–EF-G–GDP structure, but removing the atomic coordinates for the sw1 region, which was poorly defined in the CHI2 state. As the CHI2 state depicts an unstructured sw1 and as it follows the CHI–EF-G–GDP state, the GTP-binding pocket of EF-G was also modeled with GDP; the cryo-EM density at 6 Å did not allow a direct assignment. The structure of the H1–EF-G–GDP state at 6.5 Å was refined based on the final model of the H1–EF-G–GDP–Pi state removing coordinates for the unresolved sw1 region and D4 of EF-G; the partial repositioning of EF-G and the unstructured sw1 region indicated the presence of GDP in EF-G D1.

**Principal component analysis of SRL dynamics.** To analyze the dynamics of the SRL (Supplementary Fig. 2d), ribosome structures from the database were chosen without and with translational GTPases bound that represent different global ribosome states and/or different states of the respective GTPase cycles: without GTPase bound (PDB 4V6D and 4V8D), with EF-G (PDB 4V5F, 4V90 4V9H, 4V9K, 4V9O, 4V9P, 4W29, 5OT7), EF-Tu (PDB 5AFI, 5UYL, 5UYM, 6WD2, 6WD8), SelB (PDB 5LZD), IF-2 (PDB 3JCJ, 6O9K), and RF3 (PDB 4V85, 4V8O). The structures from the database and present structures (C state, H1–EF-G–GDP–Pi, and CHI1–EF-G–GDP) were aligned to 23S rRNA with pruning (cut-off 2 Å) in ChimeraX 1.2[53]. Subsequently, the SRL regions (O2′ ribose atoms of residues A2657-A2665 of 23S rRNA) of the aligned structures were subjected to principal component analysis using GROMACS 2018.8 (ref. [59]).

**Reporting summary.** Further information on research design is available in the Nature Research Reporting Summary linked to this article.

## Data availability

The cryo-EM maps/associated coordinates of atomic models generated in this study have been deposited in the Electron Microscopy Data Bank/Protein Data Bank under the following accession codes: EMD-13458 and 7PJS (C state); EMD-13459 and 7PJT (H1); EMD-13460 and 7PJU (H2); EMD-13461 and 7PJV (H1–EF-G–GDP-Pi); EMD-13462 and 7PJW (H2–EF-G–GDP-Pi); EMD-13463 and 7PJX (H1–EF-G–GDP); EMD-13464 and 7PJY (CHI1–EF-G–GDP); EMD-13465 and 7PJZ (CHI2–EF-G–GDP). The cryo-EM micrographs and particle images generated in this study have been deposited in the EMPIAR database with accession code EMPIAR-10792. Raw data of the kinetic analysis (Fig. 2a and Supplementary Fig. 2a–d) generated in this study are provided in the Source data file. The atomic coordinates used in this study are available in the Protein Data Bank under accession codes: 3J9Z; 3JCJ; 4AQY; 4V5F; 4V6D; 4V85; 4V8D; 4V8O; 4V90;

4V9H; 4V9K; 4V9O; 4V9P; 4W29; 5AFI; 5LZD; 5OT7; 5UYL; 5UYM; 5YUM; 6O9K; 6WD2; 6WD6; 6WD8; 6YSS. The authors declare that all other data supporting the findings of this study are available within the paper and its Supplementary information files. Source data are provided with this paper.

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

## Acknowledgements

We thank Holger Stark for support and encouragement, Ingo Wohlgemuth and Piotr Neumann for suggestions, Lars V. Bock for help with PCA analysis, Olaf Geintzer, Vanessa Herold, Tessa Hübner, Franziska Hummel, Sandra Kappler, Christina Kothe, Anna Pfeifer, Theresia Steiger, and Michael Zimmermann for expert technical assistance and Mario Lüttich and Tobias Koske for support in high-performance computation. This research was supported by the German Science Foundation (Deutsche Forschungsgemeinschaft, DFG) through Leibniz Prize to M.V.R. and SFB860 [collaboration between project groups A3 (M.V.R.) and A5 (Holger Stark)] and by the Max Planck Society.

## Author contributions

V.P., A.C.d.A.P.S., and N.F. conducted cryo-EM experiments and analyzed the structural data. B.-Z.P. and F.P. performed biochemical and rapid kinetics experiments and analysis. N.F. and F.P. initiated and supervised the project. N.F., F.P., and M.V.R. conceived experiments. N.F. and M.V.R. wrote the manuscript with input from all authors.

## Funding

## Competing interests

The authors declare no competing interests.
