## [Peer Review File · Nature Communications]

Structural mechanism of GTPase-powered ribosome-tRNA movementReviewers' Comments:

Reviewer #1:

Remarks to the Author:

The manuscript by Petrychenko V. et al. "Structural mechanism of GTPase-powered ribosome-tRNA movement" reports eight 70S ribosome structures bound to EF-G captured at the onset of tRNA-mRNA translocation using time-resolve cryo-EM. To prepare the complexes for cryo-EM, the components were mixed on ice to slow down the translocation reaction and EM grids frozen within 10 seconds. Using rapid kinetics under the same experimental conditions, the authors show that GTP hydrolysis occurs but Pi release is delayed. The structures differ by the state of tRNA binding, ratcheting of the small subunit body, swiveling of the 30S subunit head domain, and the position of EF-G on the ribosome. One of the reconstruction at 3.1Å resolution allows to visualize the density for GDP and Pi in the nucleotide binding pocket. The authors observe a new conformation of the N-terminal region of switch 1 (sw1). This novel fold of the sw1 region contacts helix 14 of the 16S rRNA. In this conformation, the authors propose that sw1 stabilizes rotated 30S subunit and a twisted conformation of the sarcin-ricin loop (SRL). Release of Pi from EF-G remodels the nucleotide binding pocket and relaxes the conformation of sw1, which loses interactions with the 30S and 50S subunits, thereby facilitating back-rotation of the small subunit body.

Based on the figures presented, it is difficult to assess the extent of rearrangement and the whole displacement of EF-G on the ribosome. In the text, Fig. 3a shows this movement in an unclear fashion. The cartoon shown in Fig. 4b suggests such movement but it is not shown clearly through structure superimposition. The accompanying movie shows the movement of EF-G better, coupled with 30S body and head movement, as it transitions from GDP-Pi to GDP bound. For instance, the observed 16° rotation of EF-G on the ribosome should be indicated.

It is unclear how the reported 70S-EF-G-GDP-Pi structure compares with the previous 70S-EF-G-GDPCP/NP structures, aside from sw1 that gets remodeled? Is the sw1 conformation the main difference? From the movie, the overall conformation of EF-G-GDP-Pi and EF-G-GTP appears similar, including the degree of ribosome ratcheting and head swiveling. Is this assumption accurate? The similarity and/or difference should be shown with structure superimposition as a supplementary figure.

Similarly, the authors should also compare the structure of EF-G-GDP reported here with that of Mace K et al. *Nucleic Acids Res.* 46: 3211-3217 (2018)? Mace K et al reported a structure of EF-G-GDP bound to the ribosome in the absence of inhibitor.

Overall, the figures are difficult to understand essentially because of their small size. Moving non-essential panels from the main text to supplementary materials may alleviate this issue.

Residue His37 in the N-terminal of sw1 is shown to stack with nucleotide A344 of 16S rRNA. Is there any mutational data on His37? How would this affect the function of EF-G? One may expect that mutation to Ala or Gly may abolish the conformational coupling between EF-G and the ribosome.

Minor issues:

Use universal ribosomal protein nomenclature, such as uS12, etc.

Line 5, page 5, "23 rRNA" should be 23S

Line 6, page 6, "thereby stabilizing intersubunit rotation."

The authors propose that sw1 stabilizes intersubunit rotation, and as such should be clear. Would be better to write "which appears to stabilize..." or "which we propose stabilizes..."

Fig. 3b, panel 1, page 7, The nucleotide bound is difficult to see. Similar comment for Fig. 2b, panel 1,

Suppl. Fig. 2b

Line 18, page 8, mistyped "wheras" should be "whereas"

Reviewer #2:

Remarks to the Author:

The translocation of the tRNA-mRNA complex on the ribosome and the exact role of EF-G in this process are among the key questions in translation control. This process has been extensively studied for several decades by biochemical and structural approaches. In this manuscript, Petrychenko et al. determined cryo-EM structures of a few 70S translocational intermediates by a deep classification of particles from an in vitro translocation reaction in the presence of an antibiotic Apr. Two of these structures are of great interest to the field of translation regulation, namely H1 and CH1. Overall, the manuscript is technically sound and provides important information that is missing or not well described in the current literatures.

Major concerns:

1. "Power stroke" vs "Brownian Ratchet". The two models have been heavily debated in the field. The authors appear to lean towards the first model. This reviewer has no objection in the adoption of either model in data interpretation. However, the authors have to be consistent. On page 10 line 12, the authors summarized "large-scale rigid-body rotation of EFG, which acts to bias thermal motions of the ribosome-tRNA complex into forward movement". In other places, e.g. page 4 line 1, "Binding of EF-G and GTP hydrolysis induces rotation of the ribosomal subunits and the concomitant tRNA movement from C to H". Since C to H also occurs spontaneously, the description of the role of EF-G (biasing the hybrid state) should be more accurate. This is just one example. I would recommend the authors to change the wording to stick with one model.
2. An antibiotic was also present in the sample. Therefore, the authors should discuss whether this decoding center located compound impact the structural dynamics of EF-G, particularly domain IV. In addition, as polyamines were also added to slow down the translocation steps, the mechanism of polyamines impacting translocation should also be described.
3. It is practically difficult to completely separate ribosomal particles into different groups (with regard to subtle difference on a factor or a bound small molecule). In theory, any 3D class would still be a mixture to some extent. At resolutions of 4 Å, 6 Å and 6.5 Å, assignment of GDP or GTP or GDP+Pi would be very difficult. A figure (maybe in the form of supplementary data) could be presented to show the local EM densities for the nucleotides in these different states.

Other comments:

1. The statement in Introduction, "but structural information on the early GTPase-driven steps is missing" is not correct. Previous studies have reported many intermediate structures with EF-G bound. Some of them were similar to CHI states, although these structures might not represent the authentic intermediates due to the presence of antibiotics or other inhibitors (ref 14), or the resolutions were limited (Ramrath et al., 2013). In fact, the 6.8-Å cryo-EM structure in Ramrath et al., 2013 was in an ap/P and pe/E state, which was similar to the CHI states here. The manuscript failed to cite and discuss that structure (Ramrath D.J., Lancaster L., Sprink T., Mielke T., Loerke J., Noller H.F., and Spahn C.M. (2013). Visualization of two transfer RNAs trapped in transit during elongation factor G-mediated translocation. Proc Natl Acad Sci U S A 110, 20964-20969.
2. On page 4, line 5-10, the conformation of apramycin was compared to that in a 3.5- Å crystal structure of the apramycin-30S complex. In fact, the previous crystal structure was of relatively low

resolution. Therefore, the comparisons in Supplementary Fig. 1g are not necessary.

3. A few conclusion sentences need justification.

(1) Page 4, line 25, "Collectively, these EF-G-bound structure and show how the chemical energy from GTP hydrolysis and Pi release translates into forward movement of the mRNA-tRNA complex".

(2) Page 6, line 25, "Pi release induces a large-scale repositioning of EF-G, global rearrangements of the ribosome, and the tRNA movement on the SSU (Fig. 3a)." Since Pi release is rate-limiting, and the authors only reported the co-occurrence between Pi release and conformational changes on EF-G/ribosome, it is impossible to assign a temporal order for these events.

4. All the figures in the manuscript contain many panels/subpanels. Most of the subpanels were not labelled. It is not reader-friendly.

Reviewer #1 - Remarks to the Authors

The manuscript by Petrychenko V. et al. "Structural mechanism of GTPase-powered ribosome-tRNA movement" reports eight 70S ribosome structures bound to EF-G captured at the onset of tRNA-mRNA translocation using time-resolve cryo-EM. To prepare the complexes for cryo-EM, the components were mixed on ice to slow down the translocation reaction and EM grids frozen within 10 seconds. Using rapid kinetics under the same experimental conditions, the authors show that GTP hydrolysis occurs but Pi release is delayed. The structures differ by the state of tRNA binding, ratcheting of the small subunit body, swiveling of the 30S subunit head domain, and the position of EF-G on the ribosome. One of the reconstruction at 3.1Å resolution allows to visualize the density for GDP and Pi in the nucleotide binding pocket. The authors observe a new conformation of the N-terminal region of switch 1 (sw1). This novel fold of the sw1 region contacts helix 14 of the 16S rRNA. In this conformation, the authors propose that sw1 stabilizes rotated 30S subunit and a twisted conformation of the sarcin-ricin loop (SRL). Release of Pi from EF-G remodels the nucleotide binding pocket and relaxes the conformation of sw1, which loses interactions with the 30S and 50S subunits, thereby facilitating back-rotation of the small subunit body.

Reply: We thank the reviewer for the positive evaluation of our work and the constructive criticism.

Reviewer #1 - Major Points

1. *Based on the figures presented, it is difficult to assess the extent of rearrangement and the whole displacement of EF-G on the ribosome. In the text, Fig. 3a shows this movement in an unclear fashion. The cartoon shown in Fig. 4b suggests such movement but it is not shown clearly through structure superimposition. The accompanying movie shows the movement of EF-G better, coupled with 30S body and head movement, as it transitions from GDP-Pi to GDP bound. For instance, the observed 16° rotation of EF-G on the ribosome should be indicated.*

Reply: We thank the referee for the suggestions. To better illustrate the displacement of EF-G on the ribosome we enlarged the initial figure (Fig. 1, former panel of Fig. 3a). Moreover, we added another figure showing the superposition of EF-G-GDP-Pi and EF-G-GDP in context of the ribosome in direct comparison (Figure 4a and SFig. 6a). To clarify the extent of rearrangements within EF-G, we now included a superposition of both EF-G states aligned by domain 1 of EF-G in Figure 5b and a plot depicting the C α -distances between the two states (SFig. 6b).

2. *It is unclear how the reported 70S-EF-G-GDP-Pi structure compares with the previous 70S-EF-G-GDPCP/NP structures, aside from sw1 that gets remodeled? Is the sw1 conformation the main difference? From the movie, the overall conformation of EF-G-GDP-Pi and EF-G-GTP appears similar, including the degree of ribosome ratcheting and head swiveling. Is this assumption accurate? The similarity and/or difference should be shown with structure superimposition as a supplementary figure.*

Reply: The conformation of the ribosome and the position of EF-G in our EF-G-GDP-Pi is dramatically different from the previously published 70S-EF-G-GDPCP/NP structures. Our 70S-EF-G-GDP-Pi structure shows an early pretranslocation (PRE) state with two tRNAs in the A and P sites on the SSU and EF-G outside the SSU decoding site. In contrast, the published 70S-EF-G-GDPCP/NP structures show the ribosome in the posttranslocation (POST) state, with tRNAs in P and E sites on the SSU (Tourigny et al. 2013 and Zhou et al. 2013), or lacking tRNAs (Pulk and Cate 2013) and EF-G protruding deep into the A site. That 70S-EF-G-GDPCP/NP structures show a POST state is not surprising, because translocation is known to occur also in the absence of GTP hydrolysis, albeit at a much lower rate and via a different pathway than with EF-G-GTP (see e.g. Belardinelli et al., 2016). Also, the degree of subunit rotation and SSU head domain swiveling are different in our and the previous 70S-EF-G-

GDPCP/NP structures. SSU body rotation angles are smaller and the SSU head swiveling is less pronounced in the POST structures obtained with GDPNP/CP than in our 70S-EF-G-GDP-Pi structure. The position of EF-G in the previous structures is very different from that in our EF-G-GDP-Pi state. However, the overall conformation of EF-G is surprisingly similar supporting our conclusion that translocation is mainly driven by a rigid-body displacement of EF-G, rather than by an interdomain motion of EF-G as previously thought. Local differences in EF-G include the described switch 1 region, the conformation of the β -hairpin in domain 2 (similar in previous structures to our EF-G-GDP state), and domain 4, which is stably bound to the A site on the SSU in the previous structures, but is highly dynamic in our 70S-EF-G-GDP-Pi structures.

To clarify the differences to the previous structures, we now included a corresponding statement in the main text (p. 3). We show the differences between our and the previous 70S-EF-G-GDPCP/NP in Supplementary Fig. 4a (former SFig. 2b), using the Tourigny et al. structure as a representative of this class of complexes. We also describe that despite the different location of EF-G on the ribosome, there are similarities in the GTP binding pocket between ours and Tourigny et al. structures (SFig 4a, former SFig 2b) and compare the different switch 1 fold in the structure by Zhou et al. (SFig. 4c, former SFig XY).

The CHI state is similar to previously published ones, except for the switch 1 region, which is resolved in our structure, contrary to the previous ones; this is described in Supplementary Fig. 8a,d. The ribosome conformation and EF-G-GTP position shown in the movie was only included for illustrative purposes, which is now explained in the movie caption in the Supplementary Information.

After submission of our work, two further studies appeared reporting on early translocation events, an advanced online paper in Nature by Rundlet et al. (doi.org/10.1038/s41586-021-03713-x) and a preprint in BiorXiv by Carbone et al. (doi.org/10.1101/2021.05.31.446434). We now include a paragraph in the Discussion describing the differences and similarities of those structures to the present work. (p. 7,8).

3. Similarly, the authors should also compare the structure of EF-G-GDP reported here with that of Mace K et al. *Nucleic Acids Res.* 46: 3211-3217 (2018)? Mace K et al reported a structure of EF-G-GDP bound to the ribosome in the absence of inhibitor.

Reply: The structure reported by Mace et al. represents a late post-translocation state with EF-G-GDP. We now included the reference in the introduction in the context describing previously published EF-G structures (page 2 line 14) and apologize for inadvertent omission. However, we feel that including a detailed comparison is out of scope of the current manuscript that focuses on early steps of translocation.

4. Overall, the figures are difficult to understand essentially because of their small size. Moving non-essential panels from the main text to supplementary materials may alleviate this issue.

Reply: We thank the referee for pointing this out. We agree that some of the figures in the initial submission may have been too small or too complex with too many subpanels, as pointed out also by reviewer #2. We now rearranged the original main figures to improve clarity by reducing the number of panels per figure and by enlarging individual panels. Specifically, we moved all panels on complex characterization from figure 1 into a new figure 2. The panels on global ribosome and tRNA changes from former figures 2 and 3 were integrated in figure 1. Moreover, we redistributed the former five Supplementary figures into now nine Supplementary figures.

Residue His37 in the N-terminal of sw1 is shown to stack with nucleotide A344 of 16S rRNA. Is there any mutational data on His37? How would this affect the function of EF-G? One may expect that mutation to Ala or Gly may abolish the conformational coupling between EF-G and the ribosome.

Reply: His37 is highly conserved supporting the importance of the observed stacking interaction. To the best of our knowledge, there is no mutational data available on His37.

Reviewer #1 - Minor Issues

1. *Use universal ribosomal protein nomenclature, such as uS12, etc.*

Reply: We have now switched to universal ribosome protein nomenclature throughout the text and figures.

2. *Line 5, page 5, "23 rRNA" should be 23S*

Reply: Corrected.

3. *Line 6, page 6, "thereby stabilizing intersubunit rotation." The authors propose that sw1 stabilizes intersubunit rotation, and as such should be clear. Would be better to write "which appears to stabilize..." or "which we propose stabilizes..."*

Reply: Changed accordingly.

4. *Fig. 3b, panel 1, page 7, The nucleotide bound is difficult to see. Similar comment for Fig. 2b, panel 1, Suppl. Fig. 2b*

Reply: For better distinction, we now changed the atomic representation of the nucleotides in both panels to ball-and-stick.

5. *Line 18, page 8, mistyped "wheras" should be "whereas"*

Reply: Corrected.

Reviewer #2 - Remarks to the Authors

The translocation of the tRNA-mRNA complex on the ribosome and the exact role of EF-G in this process are among the key questions in translation control. This process has been extensively studied for several decades by biochemical and structural approaches. In this manuscript, Petrychenko et al. determined cryo-EM structures of a few 70S translocational intermediates by a deep classification of particles from an in vitro translocation reaction in the presence of an antibiotic Apr. Two of these structures are of great interest to the field of translation regulation, namely H1 and CH1. Overall, the manuscript is technically sound and provides important information that is missing or not well described in the current literatures.

Reply: We thank the reviewer for the positive evaluation of our work and his helpful suggestions.

Reviewer #2 - Major Concerns

1. *"Power stroke" vs "Brownian Ratchet". The two models have been heavily debated in the field. The authors appear to lean towards the first model. This reviewer has no objection in the adoption of either model in data interpretation. However, the authors have to be consistent. On page 10 line 12, the authors summarized "large-scale rigid-body rotation of EFG, which acts to bias thermal motions of the ribosome-tRNA complex into forward movement". In other places, e.g. page 4 line 1, "Binding of EF-G*

and GTP hydrolysis induces rotation of the ribosomal subunits and the concomitant tRNA movement from C to H". Since C to H also occurs spontaneously, the description of the role of EF-G (biasing the hybrid state) should be more accurate. This is just one example. I would recommend the authors to change the wording to stick with one model.

Reply: We think that the ribosome is a Brownian machine and EF-G facilitates translocation by rectifying Brownian motions. We note that Brownian motors can generate force (Asturman, 1997; Keller and Bustamante, 2000), so there is no contradiction between force generation and a notion of Brownian machine. The statement on former p. 10 cited by the referee reflects well what we think EF-G does. We went through the manuscript to identify sentences that might be misunderstood and corrected them accordingly (p. 1, lines 22.-25). Concerning the sentence "Binding of EF-G and GTP hydrolysis induces rotation of the ribosomal subunits and the concomitant tRNA movement from C to H", we replaced 'induces' by 'facilitates', but we note that EF-G does not merely stabilize the H state that forms spontaneously. EF-G binding accelerates the forward rotation from C to H state (Sharma et al, 2016), which shows that EF-G has an active role in this process. This, however, does not mean that EF-G makes a power stroke at this point. Rather, it likely prevents rapid backward fluctuations from the rotated to non-rotated state, which in macrotime appears as active acceleration.

2. *An antibiotic was also present in the sample. Therefore, the authors should discuss whether this decoding center located compound impact the structural dynamics of EF-G, particularly domain IV. In addition, as polyamines were also added to slow down the translocation steps, the mechanism of polyamines impacting translocation should also be described.*

Reply: Our biochemical (Fig. 2 and Supplementary Fig. 2) and structural data suggest that Apr does not affect early stages of translocation, up to the CHI state formation, but blocks further progression of the tRNAs. This notion is supported by the similarities between our structures with Apr and the reported structures obtained in the absence of antibiotics, e.g. the CHI state (Zhou et al. 2014), as well as the PRE-EF-G-GDP-Pi structure by Carbone et al. that has just appeared on bioRxiv (doi.org/10.1101/2021.05.31.446434). Apr stabilizes the closed structure of the decoding center, in particular the flipped-in positions of A1492/A1493, thereby inhibiting further translocation on the SSU head domain.

Polyamines generally slow down translocation, as shown in Supplementary Fig. 2a. However, most of the structural work on translocation (and ribosome in general) was carried out in the presence of polyamines. Biochemical experiments can be done with [Ermolenko et al., RNA (2013); Kim et al., PNAS (2014); Wasserman et al., NSMB (2016)] or without [Belardinelli et al., NSMB (2016); Pan et al., Mol Cell (2007)] polyamines; there seems to be no systematic trend due to polyamines.

3. *It is practically difficult to completely separate ribosomal particles into different groups (with regard to subtle difference on a factor or a bound small molecule). In theory, any 3D class would still be a mixture to some extent. At resolutions of 4 Å, 6 Å and 6.5 Å, assignment of GDP or GTP or GDP+Pi would be very difficult. A figure (maybe in the form of supplementary data) could be presented to show the local EM densities for the nucleotides in these different states.*

Reply: The nucleotide density in H2-EF-G-GDP-Pi at 4 Å resolution, which we now show in Supplementary Fig. 1d, is compatible with both GTP and GDP-Pi. The assignment of the complex as H2-EF-G-GDP-Pi was based on its overall similarity to the earlier H1-EF-G-GDP-Pi intermediate (except for the exact position of the A-site tRNA's elbow); the latter obtained at a much higher resolution showed clear density for GDP-Pi. This is now explained in Methods, p. 13 lines 13-16. The 6 Å cryo-EM map of the CHI2-EF-G-GDP state did not allow a direct nucleotide assignment. However,

the CHI2 state follows the CHI1–EF–G–GDP state and depicts an unstructured sw1. Therefore, the G domain was also modelled with GDP (p. 13 lines 21-23). In the off-pathway H1–EF–G–GDP (Supplementary Fig. 3), the density for the nucleotide is not well resolved, but because EF–G is partially repositioned and the switch 1 region is unstructured, as in the CHI–EF–G–GDP structures, we assume that the GDP-form is more likely (p. 13 lines 25-26).

Reviewer #2 - Other comments:

1. *The statement in Introduction, “but structural information on the early GTPase-driven steps is missing” is not correct. Previous studies have reported many intermediate structures with EF–G bound. Some of them were similar to CHI states, although these structures might not represent the authentic intermediates due to the presence of antibiotics or other inhibitors (ref 14), or the resolutions were limited (Ramrath et al., 2013). In fact, the 6.8-Å cryo-EM structure in Ramrath et al., 2013 was in an ap/P and pe/E state, which was similar to the CHI states here. The manuscript failed to cite and discuss that structure (Ramrath D.J., Lancaster L., Sprink T., Mielke T., Loerke J., Noller H.F., and Spahn C.M. (2013). Visualization of two transfer RNAs trapped in transit during elongation factor G-mediated translocation. Proc Natl Acad Sci U S A 110, 20964-20969.*

Reply: We now included the citation in the paper (page 2, line 14 and page 4, lines 2,4). This is certainly an important paper which was the first one to identify the CHI states, albeit at a low resolution.

2. *On page 4, line 5-10, the conformation of apramycin was compared to that in a 3.5- Å crystal structure of the apramycin-30S complex. In fact, the previous crystal structure was of relatively low resolution. Therefore, the comparisons in Supplementary Fig. 1g are not necessary.*

Reply: The respective panel was removed from the Supplementary Figures.

3. *A few conclusion sentences need justification.*

(1) Page 4, line 25, “Collectively, these EF–G-bound structure and show how the chemical energy from GTP hydrolysis and Pi release translates into forward movement of the mRNA–tRNA complex”.

Reply: Previous biochemical experiments have shown that Pi release triggers SSU unlocking and tRNA translocation on the SSU [Savelsbergh et al., Mol. Cell (2003)]. SSU unlocking is the major barrier to overcome in translocation, which that is extremely slow in the absence of EF–G and slow in the absence of GTP hydrolysis [Katunin et al., Biochemistry (2002); Belardinelli et al., NSMB (2016)]. Solving the translocation intermediates before and after Pi release reveals the mechanism of SSU unlocking. While this is described in the Introduction, we added a short sentence on p. 4, lines 9-11 to explain the importance of the comparison.

(2) Page 6, line 25, “Pi release induces a large-scale repositioning of EF–G, global rearrangements of the ribosome, and the tRNA movement on the SSU (Fig. 3a).” Since Pi release is rate-limiting, and the authors only reported the co-occurrence between Pi release and conformational changes on EF–G/ribosome, it is impossible to assign a temporal order for these events.

Reply: We do not imply the order of events on the ribosome, but still think that the changes are triggered by Pi release, in a way similar to all GTPases where the release of Pi triggers conformational rearrangements that terminate signaling [Vetter and Wittinghofer, Science (2001)]. We changed “induces” to “triggers”, which we think does not have an (unintended) temporal order. The similarities of EF–G to “conventional” GTPases like EF–Tu are explained in the last paragraph of the Discussion (p. 8).

4. *All the figures in the manuscript contain many panels/subpanels. Most of the subpanels were not labelled. It is not reader-friendly.*

Reply: We agree and rearranged the figures to reduce the number of panels (Compare also our reply to major points 1 and 4 by Reviewer #1). Subpanels in Figs. 3 and 4 are labeled with numbers that correspond to EF-G regions shown in the first panel of the respective Fig. We also checked that Fig. legends contain all necessary explanations and hope that these changes improved the clarity of the Figs. and the readability of the paper.

Reviewers' Comments:

Reviewer #1:

Remarks to the Author:

The authors have addressed this reviewer's concerns.

Minor points:

Abstract, page 1, line 9: "pivoting at the sarcin-ricin loop..." better as "pivoting around the..."

Page 3, line 1, "We obtained eight main structures..." The issue here is that Fig. 1a shows seven structures. The reader has to go to Suppl. Fig. 1 to realize that main text Fig. 1 lacks the H1-EF-G-GDP structure. This reviewer understands that H1-EF-G-GDP may be an unproductive state captured (as alluded by the authors on page 4, lines 7-9), and then it is fine to show only the seven main structures in Fig. 1a, but the text should be changed accordingly to "seven main structures".

Page 4, line 23: "GTPsensor" missing a space

Page 7, lines 25-26, "is similar to our and the Carbone et al. CHI-EF-G-GDP complex, except for the switch 1 region, which is not resolved in that structure." It brings confusion as the authors wrote above in lines 15-16 "except for the sw1 region which is not resolved in Carbone et al." It could be clarified by "is similar to our and the Carbone et al. CHI-EF-G-GDP complex, except for the switch 1 region, which is not resolved in Rundlet et al. and Carbone et al. structures."

Page 9, line 30, "BPB" should be PBP.

Suppl. Information, Page 9, line 13, "...this paper..." better as "...this study". Same comment elsewhere in the main text.

Reviewer #2:

Remarks to the Author:

The authors have addressed most of my concerns. In the response letter, the authors cited published papers to support their assignment of different nucleotide states. I understand that it is difficult to make unambiguous assignment solely based on structural data. Nevertheless, I would suggest the authors to soften the tone a little bit in the results. Overall, I think the manuscript is ready for publication as long as this is revised.

We thank the reviewers for excellent suggests and the help in improving our manuscript. We implemented all their suggestions and used the occasion to proofread the manuscript one more time and remove residual typos and grammar problems.

Reviewer #1 - Remarks to the Authors

The authors have addressed this reviewer's concerns.

Reviewer #1 - Minor Issues

1. Abstract, page 1, line 9: "pivoting at the sarcin-ricin loop..." better as "pivoting around the..."

Reply: Done, thank you.

2. Page 3, line 1, "We obtained eight main structures..." The issue here is that Fig. 1a shows seven structures. The reader has to go to Suppl. Fig. 1 to realize that main text Fig. 1 lacks the H1-EF-G-GDP structure. This reviewer understands that H1-EF-G-GDP may be an unproductive state captured (as alluded by the authors on page 4, lines 7-9), and then it is fine to show only the seven main structures in Fig. 1a, but the text should be changed accordingly to "seven main structures".

Reply: Changed as suggested (due to this change we also swapped the order of Supplementary Figs. 3 and 4).

3. Page 4, line 23: "GTPsensor" missing a space.

Reply: Done, thank you.

4. Page 7, lines 25-26, "is similar to our and the Carbone et al. CHI-EF-G-GDP complex, except for the switch 1 region, which is not resolved in that structure." It brings confusion as the authors wrote above in lines 15-16 "except for the sw1 region which is not resolved in Carbone et al." It could be clarified by "is similar to our and the Carbone et al. CHI-EF-G-GDP complex, except for the switch 1 region, which is not resolved in Rundlet et al. and Carbone et al. structures."

Reply: Changed as suggested. A few additional minor corrections were introduced in the same paragraph to remove redundancies in the text and improve grammar.

5. Page 9, line 30, "BPB" should be PBP.

Reply: Done, thank you.

6. Suppl. Information, Page 9, line 13, "...this paper..." better as "...this study". Same comment elsewhere in the main text.

Reply: Changed in the main text and Supplementary.

Reviewer #2 - Remarks to the Authors

The authors have addressed most of my concerns. In the response letter, the authors cited published papers to support their assignment of different nucleotide states. I understand that it is difficult to make unambiguous assignment solely based on structural data. Nevertheless, I would suggest the authors to soften the tone a little bit in the results. Overall, I think the manuscript is ready for publication as long as this is revised.

Reply: To address this question, a sentence has been introduced on p. 3 to indicate that the assignment of the H2-EF-G-GDP-Pi structure is based on both cryo-EM densities and biochemical data of Fig. 2a:

'In the lower resolution H2-EF-G-GDP-Pi state, the nucleotide density would be compatible with either GTP or GDP-Pi (Supplementary Figure 1d). However biochemical data showing that GTP has been hydrolyzed at this time point of cryo-EM preparation (Fig. 2a) and the overall structure similarity to the H1-EF-G-GDP-Pi complex suggests the presence of GDP-Pi, rather than GTP, also in the H2 state with EF-G.'

A very similar sentence has been delete from Methods to avoid redundancy.